# New Pyrroline Isolated from Antarctic Krill-Derived Actinomycetes *Nocardiopsis* sp. LX-1 Combining with Molecular Networking

**DOI:** 10.3390/md21020127

**Published:** 2023-02-15

**Authors:** Ting Shi, Yan-Jing Li, Ze-Min Wang, Yi-Fei Wang, Bo Wang, Da-Yong Shi

**Affiliations:** 1College of Chemical and Biological Engineering, Shandong University of Science and Technology, Qingdao 266590, China; 2State Key Laboratory of Microbial Technology, Institute of Microbial Technology, Shandong University, Qingdao 266237, China

**Keywords:** molecular networking, Antarctic krill, *Nocardiopsis* sp. LX-1, pyrroline, nocarpyrroline A, antimicrobial activity

## Abstract

Antarctic krill (*Euphausia superba*) of the *Euphausiidae* family comprise one of the largest biomasses in the world and play a key role in the Antarctic marine ecosystem. However, the study of *E. superba*-derived microbes and their secondary metabolites has been limited. Chemical investigation of the secondary metabolites of the actinomycetes *Nocardiopsis* sp. LX-1 (in the family of *Nocardiopsaceae*), isolated from *E. superba*, combined with molecular networking, led to the identification of 16 compounds **a**–**p** (purple nodes in the molecular network) and the isolation of one new pyrroline, nocarpyrroline A (**1**), along with 11 known compounds **2**–**12**. The structure of the new compound **1** was elucidated by extensive spectroscopic investigation. Compound **2** exhibited broad-spectrum antibacterial activities against *A. hydrophila*, *D. chrysanthemi*, *C. terrigena*, *X. citri pv. malvacearum* and antifungal activity against *C. albicans* in a conventional broth dilution assay. The positive control was ciprofloxacin with the MIC values of <0.024 µM, 0.39 µM, 0.39 µM, 0.39 µM, and 0.20 µM, respectively. Compound **1** and compounds **7**, **10**, and **11** displayed antifungal activities against *F. fujikuroi* and *D. citri*, respectively, in modified agar diffusion test. Prochloraz was used as positive control and showed the inhibition zone radius of 17 mm and 15 mm against *F. fujikuroi* and *D. citri*, respectively. All the annotated compounds **a**–**p** by molecular networking were first discovered from the genus *Nocardiopsis*. Nocarpyrroline A (**1**) features an unprecedented 4,5-dihydro-pyrrole-2-carbonitrile substructure, and it is the first pyrroline isolated from the genus *Nocardiopsis*. This study further demonstrated the guiding significance of molecular networking in the research of microbial secondary metabolites.

## 1. Introduction

The extreme environments of Antarctica, including severe cold, an arid climate, and solar radiation, have created a unique ecological system. Creatures in the Antarctic ecosystem, in particular microorganisms, usually have to produce some structurally specific active substances to adapt to the harsh conditions [1]. Antarctic krill (*Euphausia superba*), a small crustacean in the family of *Euphausiidae* in the Antarctic Ocean, with one of the largest biomasses (approximately 379 million metric tons) in the world, plays a key role in the Antarctic marine ecosystem [2]. Antarctic krill is critical in the food chain for seals, whales, and penguins, making it the foundation of the Southern Ocean ecosystem and an important marine living resource [3,4]. The research on Antarctic krill has mainly focused on its fisheries [5], nutritive value [6], and distribution [7]. To the best of our knowledge, only three studies have focused on Antarctic krill-derived microbes and their secondary metabolites [8,9,10], so there is great potential to discover new bioactive natural products from *Euphausia superba* symbiotic microorganisms.

The genus *Nocardiopsis* is one of the important actinomyces for its widespread application in industry [11], agriculture [12], and environmental protection [13], and especially for its ability to produce a wide variety of new compounds with different skeletons and diverse biological activities [14,15,16]. The bioactive compounds produced by *Nocardiopsis* can assist it to survive under different and hostile environmental conditions, even in the Antarctic marine ecosystem [14,17].

Molecular networking has been an excellent methodology by which to visualize and annotate the secondary metabolites of fungi crude extracts based on nontargeted tandem mass (MS/MS) data in the Global Natural Products Social Molecular Networking (GNPS; http://gnps.ucsd.edu) platform [18,19,20]. Molecular networking has become an efficient way to dereplicate mixtures and discover new natural products [21]. In our ongoing research of secondary metabolites produced by Antarctic microorganisms, some new bioactive compounds have been found with molecular networking [22,23].

Antimicrobial resistance (AMR) has become an increasing problem with the misuse and overuse of antibiotics, causing infections increasingly difficult or impossible to treat with existing drugs [24,25,26]. The WHO has declared AMR to be one of the top 10 global public health threats. Antibiotic shortages are affecting countries in all regions. A lack of access to quality antimicrobials also remains a major issue [27]. Natural products from microorganisms have played a significant role in delivering antibiotics since the discovery of penicillin in the 1940s [28].

Chemical investigation of the fermentation broth of crude extracts of the actinomycetes *Nocardiopsis* sp. LX-1, derived from the Antarctic krill *E. superba*, has been carried out to find more antimicrobial compounds. The antimicrobial activityof the crude extracts against *Candida albicans* showed an inhibition rate of 70 ± 0.46% at the concentration of 100 µg/mL, combined with molecular networking. In this report, the compounds recognized by molecular networking and the isolation, structure elucidation, and antimicrobial activity evaluation of the isolated compounds are discussed.

## 2. Results and Discussion

### 2.1. Visualized Secondary Metabolic Profile and Identified Compounds by Molecular Networking

The actinomycetes *Nocardiopsis* sp. LX-1 was cultured in nutrient broth (NB) and extracted by EtOAc repeatedly to obtain organic crude extracts. Then, the crude extract samples were diluted and subjected to UHPLC-MS/MS analysis to obtain the raw MS/MS data. The obtained MS/MS data were converted into .mzXML format and uploaded to the GNPS platform to obtain the molecular network (Figure 1, https://gnps.ucsd.edu/ProteoSAFe/status.jsp?task=37dd96194c924f6d9daeef62672ba930, accessed on 29 November 2021) of the secondary metabolic profile of the actinomycetes *Nocardiopsis* sp. LX-1. This was finally visualized by Cytoscape software. The molecular network of the metabolic profile of LX-1 contained 473 nodes and 530 edges, meaning there were 473 compounds annotated by molecular networking in the raw MS/MS data of the actinomycetes crude extracts. The compounds that shared similar MS/MS fragments and similar chemical structures were connected by edges and clustered together into a molecular family. There were 31 molecular clusters (over two nodes) in the LX-1 molecular network, and the maximal cluster contained 33 nodes, meaning 33 compounds with similar chemical structures were in the first molecular cluster (Figure 1). The purple nodes in Figure 1 express compounds annotated by molecular networking and blue nodes mean compounds not identified by molecular networking. Most of the compounds (blue nodes) in the molecular network were unknown after searching in the GNPS database (Figure 1), indicating the crude extracts produced by *Nocardiopsis* sp. LX-1 might possess abundant new compounds.

The molecular networking was used to preliminarily investigate the number and the structural types of the secondary metabolites of the actinomycetes *Nocardiopsis* sp. LX-1 and predicted the possibility of finding new compounds to determine the research value of the actinomycetes. After analysis by molecular networking, 16 compounds **a**–**p** (purple nodes) (Figure 2 and Figure 3, Table 1) were characterized from the crude extracts of *Nocardiopsis* sp. LX-1 by comparing their MS/MS spectra with those in the GNPS library. The plastic products were avoided during the extraction and purification, and the plasticizers **i** and **j** might be contaminated by the organic extracted solvent, which were not the LC-MS grade/HPLC grade solvents that might contain contaminants. All the compounds **a**–**p**, identified by molecular networking, were preliminarily detected from the crude extracts of *Nocardiopsis* sp. LX-1. The big cluster families of the LX-1 molecular network with heavy molecular weight are speculated to be fatty acid compounds which could be lost in separated process. A series of antimicrobial isoflavonoids and flavonoids (**a**–**f**) were identified by molecular networking, which should be the target natural products to be isolated from LX-1. None of the recognized compounds **a**–**p** had been isolated from the genus *Nocardiopsis* until now, indicating the ability of molecular networking to analyze microbial secondary metabolites and guide the directional separation.

### 2.2. Structure Elucidation and Antimicrobial Activity of Isolated Compounds **1**–**12**

Chemical investigation of the broth fermentative crude extracts of *Nocardiopsis* sp. LX-1 was carried out to find the molecular networking analyzed compounds, and led to the isolation of one new compound, nocarpyrroline A (**1**), along with 11 known compounds **2**–**12** (Figure 4). However, only one target flavonoid derivative, daidzein (**2**, same as the molecular networking identified isoflavonoid **e**), in the molecular network was isolated, which might be due to the insufficient fermentation of *Nocardiopsis* sp. LX-1 and its low yield of the isoflavonoid and flavonoid compounds. This is the first time isoflavonoid has been isolated from the genus *Nocardiopsis*. Compounds **1**, **2**, and **4**–**8** were appeared in the molecular network of *Nocardiopsis* sp. LX-1 as individual nodes (Figure 5). The self-loops of compounds **1** and **2** in the network might because their MS/MS spectra are not informative due to the low amount of the compounds; therefore no peak matches with other clustered nodes. Compounds **4**–**8** were appeared as individual nodes might be due to the fact that cyclic dipeptides are harder to be ionized in the positive ESI MS/MS experiment than in the negative ESI MS/MS experiment. Compound **9** was not detected in ESI^+^ MS/MS spectra of *Nocardiopsis* sp. LX-1 because it cannot ionize in positive MS/MS measurement. Compounds **8** and **10**–**12** were not found in the molecular network might be due to their low molecular weight, which is hard to be detect.

Nocarpyrroline A (**1**) was obtained as an amorphous white powder. Its molecular formula was determined as C_14_H_14_O_3_N_2_ by HR-ESI-MS with the [M−H]^−^ peak at *m/z* 257.0939 (calculated for C_14_H_13_O_3_N_2_, 257.0932) (Appendix A) and [M + H]^+^ peak at *m/z* 259.1080 (calculated for C_14_H_15_O_3_N_2_, 259.1077) (Appendix A), containing nine degrees of unsaturation. The five aromatic proton signals at *δ*_H_ 7.09–7.04 (1H, m), 7.24–7.20 (1H, m), 7.20–7.17 (1H, m), 7.24–7.20 (1H, m), and 7.09–7.04 (1H, m) in ^1^H NMR data of **1** (Table 2 and Appendix A), combined with the six ^13^C NMR signals at *δ*_H_ 135.6 C, 131.0 CH, 129.3 CH, 128.2 CH, 129.3 CH, 131.0 CH (Table 2 and Appendix A), indicated that there was a phenyl group in **1**. The HMBC correlations from H-8 to C-9/C-10/C-14 proved the phenyl group was linked at C-8 (Figure 6). The ^1^H NMR, ^13^C NMR, HSQC and HR-ESI-MS spectra (Appendix A) of **1** displayed one unsaturated quaternary carbon at *δ*_C_ 135.2, one unsaturated methine at *δ*_H_ 5.73 (1H, d, 2.7 Hz), *δ*_C_ 119.6, one oxygenated methine at *δ*_H_ 4.71 (1H, ddd, 8.2, 3.5, 2.7 Hz), *δ*_C_ 70.2, and one methylene at *δ*_H_ 3.72 (1H, dd, 13.6, 8.2 Hz), 3.62 (1H, dd, 13.6, 3.5 Hz), and *δ*_C_ 54.5, demonstrated a pyrroline ring with a hydroxyl substituent group in **1**. The COSY cross-peaks of H-3/H-4 and H-4/H-5, and the HMBC correlations from H-3 to C-2 and H-3 to C-5 further proved the existence of a pyrroline ring in **1** (Figure 6). One unsaturated quaternary carbon at *δ*_C_ 164.2 showed there was a carbonyl group in **1**. The HMBC correlation between H-8 and C-6, and the COSY cross peak of H-7/H-8, revealed the carbonyl group was in the location of C-6 (Figure 6). The high field shift of the C-6 carbonyl group indicated the amide linkage between C-6 and N-1 (Table 2). The IR absorption band at 2253 cm^–1^ (Appendix A), combined with the molecular formula of C_14_H_14_O_3_N_2_ and the nine degrees of unsaturation suggested that there was one cyano group in **1**. The high field shift of C-2 and low field shift of C-3 indicated that the cyano group was linked at C-2 (Table 2). Thus, the plane structure of **1** was established unambiguously as shown in Figure 4. The NOESY spectrum was measured to determine the relative configuration of **1**; however, the NOESY cross-peaks were not clear enough to identify the relative configuration of **1**. The absolute configuration of **1** was attempted to determine by the modified Mosher’s method. Unfortunately, it was failed due to the limited quantity of **1**.

The new compound nocarpyrroline A (**1**) was proposed to be biosynthesized by the condensation reaction of hydroxyl-cinnamate and 4-hydroxy-4,5-dihydro-1*H*-pyrrole-2-carbonitrile (Figure 7). Hydroxyl-cinnamate was suggested to be obtained from the hydroxylation of cinnamate. Phenyllactic acid (PLA) was biosynthesized from phosphoenolpyruvate and erythrose-4-phosphate through the enzymes of 3-deoxy-7-phosphoheptulonate synthase (DAHPS), 3-dehydroquinic acid synthase (DHQS), 3-dehydroquinic acid dehydratase (DHQD), shikimic acid 5-dehydrogenase (SDH), shikimic acid kinase (SK), 3-enolpyruvylshikimic acid 5-phosphate synthase (ESPS), chorismic acid synthase (CS), chorismic acid mutase (CM), prephenic acid aminotransferase (PAT), arogenic acid dehydratase (ADT), aminotransferase (ATF), dehydrogenase (DHG) in sequence [29,30]. Then, 4-hydroxy-4,5-dihydro-1*H*-pyrrole-2-carbonitrile was deduced to be achieved through amination, dehydration, hydroxylation, and reduction of L-Pro. L-Pro was biosynthesized from L-glutamate-*γ*-semialdehyde catalyzed by Δ^1^-pyrroline-5-carboxylate reductase. L-glutamate-*γ*-semialdehyde could be acquired by Δ^1^-pyrroline-5-carboxylate synthase from L-Glu or received through catalyzing L-ornithine by ornithine-*δ*-aminotransferase (Figure 7) [31].

According to the plausible biogenetic pathway proposed for **1** (Figure 7), the 4-OH was deduced to be the *R* configuration to avoid steric hindrance. There were 75 conformers of **1**-4*R*,7*S* with the minimum energy of 322.98 kJ/mol and 79 conformers of **1**-4*R*,7*R* with the minimum energy of 326.88 kJ/mol through molecular mechanics MMFF method, so compound **1** tend to be 4*R*,7*S* with lower energy. The phenyllactic acid (PLA) part of **1** can be biosynthesized as *L*-PLA by *L*-form dehydrogenase (DHG) or as *R*-PLA by *R*-form DHG (Figure 7) [29]. PLA has been found to exhibit antimicrobial activities including a range of Gram-positive bacteria, Gram-negative bacteria, yeasts and mould species [29]. Compared with **1**, only two compounds—phenylmethyl 1-acetyl-4,5-dihydro-4-hydroxy-1H-pyrrole-2-carboxylate (CAS number: 2293990-67-5) [32] and 1-(2,3-dihydro-3,3,5-trimethyl-1H-pyrrol-1-yl)-2-phenoxyethanone (CAS number: 118428-79-8) [33]—show structural similarity with over 80% searched in SciFinder, and both of the compounds were obtained through chemical synthesis. Compound **1** appeared alone in the molecular network of *Nocardiopsis* sp. LX-1 (Figure 5), indicating there is no compound showing enough MS/MS similarity with the single node in this certain sample.

Compound **2** was obtained as an amorphous white powder. Its molecular formula was determined as C_15_H_10_O_4_ by HR-ESI-MS with the [M − H]^−^ peak at *m/z* 253.0501 (calculated for C_15_H_9_O_4_, 253.0506) (Appendix A) and [M + H]^+^ peak at *m/z* 255.0660 (calculated for C_15_H_11_O_4_, 255.0652) (Appendix A). The 15 unsaturated carbon signals at *δ*_C_ 152.4, 123.3, 174.6, 127.0, 115.0, 158.9, 102.0, 158.0, 116.5, 122.9, 130.1, 115.0, 157.0, 115.0 and 130.1, as well as eight unsaturated hydrogen signals at *δ*_H_ 8.20 (1H, s), 7.87 (1H, d, 8.8), 6.82 (1H, dd, 8.8, 2.2), 6.69 (1H, d, 2.2), 7.36 (1H, d, 8.6), 6.79 (1H, d, 8.6), 6.79 (1H, d, 8.6), and 7.36 (1H, d, 8.6) indicated that **2** was an isoflavonoid. Compound **2** was further determined as daidzene for its nearly identical ^1^H NMR and ^13^C NMR data (Appendix A), as compared to examples in the literature [34].

Compound **3** was obtained as an amorphous white powder. The HR-ESI-MS spectra (Appendix A) of **3** exhibited its molecular formula as C_10_H_16_O_2_N_2_ with the [M − H]^−^ peak at *m/z* 195.1130 (calculated for C_10_H_15_O_2_N_2_, 195.1139) and [M + H]^+^ peak at *m/z* 197.1294 (calculated for C_10_H_17_O_2_N_2_, 197.1285). The two amido-carbonyl signals at *δ*_C_ 169.9 and 165.5, and two amino-methine signals at *δ*_H_ 4.07, dd (10.1, 5.6), *δ*_C_ 58.4 and *δ*_H_ 3.70–3.62, m, *δ*_C_ 63.5 in NMR data proved that compound **3** was a diketopiperazine (DKP) (Appendix A). Further analysis showed that the ^1^H NMR, ^13^C NMR, and specific optical rotation (OR) data (Appendix A) of **3** demonstrated that **3** was cyclo(d-Pro-l-Val) [35].

Compound **4** was acquired as an amorphous white powder. Its molecular formula was decided by HR-ESI-MS spectra (Appendix A) as C_11_H_18_O_2_N_2_ with the [M − H]^−^ peak at *m/z* 209.1288 (calculated for C_11_H_17_O_2_N_2_, 209.1296) and [M + H]^+^ peak at *m/z* 211.1449 (calculated for C_11_H_19_O_2_N_2_, 211.1441), which was similar with those of **3**. It was deduced that **4** was a DKP compound similar to **3**. This deduction was further confirmed by the fact that the NMR data of **4** displayed two amido-carbonyl signals at *δ*_C_ 169.8 and 165.5, and two amino-methine signals at *δ*_H_ 4.07, dd (9.9, 6.4), *δ*_C_ 58.5 and *δ*_H_ 3.77, dd (5.7, 3.9), *δ*_C_ 62.9 (Appendix A). Compound **4** was finally proved to be cyclo(4-methyl-d-Pro-l-Nva) (**4**) Its ^1^H NMR, ^13^C NMR, and specific OR data (Appendix A) are almost identical to examples in the literature [36].

Compound **5** was isolated as a colorless oil. The molecular formula of **5** was ascertained as C_14_H_16_O_3_N_2_ through HR-ESI-MS spectra (Appendix A) with the [M − H]^−^ peak at *m/z* 259.1083 (calculated for C_14_H_15_O_3_N_2_, 259.1088), and [M + H]^+^ peak at *m/z* 261.1244 (calculated for C_14_H_17_O_3_N_2_, 261.1234) and 283.1063 (calculated for C_14_H_16_O_3_N_2_Na, 283.1053). The NMR spectra data of two amido-carbonyl signals at *δ*_C_ 166.9 and 171.0, and two amino-methine signals at *δ*_H_ 4.33, ddd (11.8, 5.9, 2.0), *δ*_C_ 58.1 and *δ*_H_ 4.45, td (5.1, 1.8), *δ*_C_ 57.4 (Appendix A) illustrated that **5** was a DKP compound. Further analysis of the ^1^H NMR, ^13^C NMR, and specific OR data (Appendix A) of **5** proved that **5** was cyclo(*trans*-4-OH-l-Pro-l-Phe) [37].

Compound **6** was obtained as a colorless oil. Its molecular formula was determined as C_14_H_16_O_2_N_2_ through HR-ESI-MS spectra (Appendix A) with the [M − H]^−^ peak at *m/z* 243.1133 (calculated for C_14_H_15_O_2_N_2_, 243.1139), and [M + H]^+^ peak at *m/z* 245.1294 (calculated for C_14_H_17_O_2_N_2_, 245.1285). The only obvious difference between the NMR data of **6** and **5** was the methylene at C-4 in **6** was substituted by hydroxy-methine in **5**. We carefully compared the NMR and specific OR data of **6** (Appendix A) with those in the literature [38] and came to the determination that **6** is cyclo(l-Pro-l-Phe).

Compound **7** was acquired as a colorless oil. The HR-ESI-MS spectra (Appendix A) of **7** exhibited the same molecular formula as **6**, with the [M − H]^−^ peak at *m/z* 243.1133 (calculated for C_14_H_15_O_2_N_2_, 243.1139), and [M + H]^+^ peak at *m/z* 245.1293 (calculated for C_14_H_17_O_2_N_2_, 245.1285) and 267.1112 (calculated for C_14_H_16_O_2_N_2_Na, 267.1104). The NMR and specific OR data (Appendix A) determined that **7** was cyclo(d-Pro-l-Phe), according to the literature [39].

Compound **8** was obtained as an amorphous white powder. The molecular formula of C_11_H_18_O_3_N_2_ was determined by HR-ESI-MS with the [M − H]^−^ peak at *m/z* 225.1238 (calculated for C_11_H_17_O_3_N_2_, 225.1245), and [M + H]^+^ peak at *m/z* 227.1398 (calculated for C_11_H_19_O_3_N_2_, 227.1390) and 249.1217 (calculated for C_11_H_18_O_3_N_2_Na, 249.1210) which was similar to those of **4**. We deduced that **8** was a DKP compound similar to **4**. This deduction was further confirmed by the NMR data of **8**, which displayed two amido-carbonyl signals at *δ*_C_ 168.8 and 172.9, and two amino-methine signals at *δ*_H_ 4.50, ddd (11.2, 6.5, 1.6), *δ*_C_ 58.5 and *δ*_H_ 4.15, ddd (6.6, 4.4, 1.8), *δ*_C_ 54.9 (Appendix A). Compound **8** was finally determined to be cyclo(4-hydroxyl-l-Pro-l-Leu) because it has almost the same ^1^H NMR, ^13^C NMR, and specific OR data (Appendix A) as examples in the literature [40].

Compound **9** was obtained as an amorphous white powder. Its molecular formula was determined to be C_8_H_6_O_2_N_2_ by HR-ESI-MS (Appendix A) with the [M − H]^−^ peak at *m/z* 161.0345 (calculated for C_8_H_5_O_2_N_2_, 161.0357). The ^1^H NMR and ^13^C NMR spectra of **9** exhibited eight unsaturated carbon signals at *δ*_C_ 151.0, 163.5, 127.5, 122.7, 135.4, 116.0, 141.8, and 114.9, and four aromatic-hydrogen signals at *δ*_H_ 7.87 (1H, dd, 8.3, 1.6), 7.17–7.13 (2H, m), and 7.62 (1H, ddd, 8.5, 7.3, 1.6) (Appendix A), which were the NMR characteristic of quinazolinedione and almost same as those in the literature [41], so **9** was identified as 2,4(1*H*, 3*H*)quinazolinedione.

Compound **10** was acquired in the form of colorless needles. The HR-ESI-MS (Appendix A) of **10** showed the molecular formula as C_4_H_4_O_2_N_2_ with the [M − H]^−^ peak at m/z 111.0187 (calculated for C_4_H_3_O_2_N_2_, 111.0200), and [M + H]^+^ peak at *m/z* 113.0351 (calculated for C_4_H_5_O_2_N_2_, 113.0346). The two aromatic quaternary carbon signals at *δ*_C_ 151.6 and 164.4, two aromatic methine signals at *δ*_H_ 5.44, dd (7.4, 1.9), *δ*_C_ 100.3 and *δ*_H_ 7.39, dd (7.4, 5.6), *δ*_C_ 142.3, and two hydrogen bond signals at 10.83, br s and 11.03, br s in NMR data (Appendix A) determined **10** to be uracil [42].

Compound **11** was obtained as a pale yellow powder. The molecular formula of C_10_H_8_O_2_ was determined by HR-APCI-MS (Appendix A) with the [M + H]^+^ peak at *m/z* 161.0602 (calculated for C_10_H_9_O_2_, 161.0597). The nine unsaturated carbon signals at *δ*_C_ 194.7, 128.1, 132.7, 131.2, 121.3, 121.9, 123.0, 111.6, and 113.5 and five aromatic-hydrogen signals at *δ*_H_ 8.18 (1H, d, 2.8), 8.21 (1H, ddd, 6.6, 2.8, 1.4), 7.24–7.18 (2H, m), 7.44 (1H, dd, 6.5, 2.3), and one hydroxy-methylene at *δ*_H_ 4.72 (2H, d, 2.8), *δ*_C_ 64.9 in NMR data (Appendix A) proved that compound **11** was salvinin A [43].

Compound **12** was obtained as a pale yellow powder. The molecular formula of C_7_H_7_O_2_N was determined by HR-ESI-MS (Appendix A) with the [M − H]^−^ peak at *m/z* 136.0393 (calculated for C_7_H_6_O_2_N, 136.0404), and [M + H]^+^ peak at *m/z* 138.0554 (calculated for C_7_H_8_O_2_N, 138.0550). The NMR spectra of **12** displayed six phenyl-carbon signals at *δ*_C_ 109.7, 151.2, 116.6, 135.2, 116.9, and 132.3, along with four phenyl-hydrogen signals at *δ*_H_ 6.67 (1H, dd, 7.1, 1.0), 7.30 (1H, ddd, 8.3, 7.1, 1.6), 6.66 (1H, dd, 8.3, 1.0), and 7.92 (1H, dd, 8.3, 1.6), and one carboxyl signal at *δ*_C_ 173.6 (Appendix A) indicated that **12** was 2-aminobenzoic acid [44].2.3. Antimicrobial Activities of Compounds **1**–**12**

All the actinomycetes *Nocardiopsis* sp. LX-1 isolated compounds (**1**–**12**) were evaluated for their antimicrobial activities. For the fermentation broth, crude extracts of the actinomycetes LX-1 displayed antimicrobial activity against *C. albicans* with the inhibition of 70 ± 0.46% at the concentration of 100 µg/mL. The antibacterial activities were tested against a panel of bacteria, including five phytopathogenic bacteria (*X. axonopodis*, *X. citri pv. malvacearum*, *D. chrysanthemi*, *P. syringae*, and *C. terrigena*), four animal pathogenic bacteria (*B. subtilis*, *E. coli*, *P. aeruginosa*, and *S. aureus*), and eight marine fouling bacteria (*A. salmonicida*, *A. hydrophila*, *E. cloacae*, *P. angustum*, *P. halotolerans*, *V. anguillarum*, *V. harveyi*, and *P. fulva*). Compounds **1**, **2**, **7**, and **10** showed antibacterial activities against *A. hydrophila* with the MIC values of 100 µM, 100 µM, 100 µM, and 50 µM, respectively. Compound **2** also exhibited antibacterial activities against *D. chrysanthemi*, *C. terrigena*, and *X. citri pv. malvacearum* with the MIC values of 100 µM, 100 µM, and 25 µM, respectively. The MIC values of the positive control ciprofloxacin (CPFX) against *A. hydrophila*, *D. chrysanthemi*, *C. terrigena*, and *X. citri pv. malvacearum* were < 0.024 µM, 0.39 µM, 0.39 µM, and 0.39 µM, respectively.

The antifungal activities of the compounds **1**–**12** were also measured against one animal pathogenic fungus (*C. albicans*) and eight phytopathogenic fungi (*A. niger*, *D. citri*, *F. fujikuroi*, *F. proliferatum*, *F. oxysporum*, *F. graminearum*, *Colletotrichum* sp., and *A. alternata*). Compound **2** displayed antifungal activity against *C. albicans* with the MIC value of 100 µM. The MIC value of the positive control CPFX against *C. albicans* was 0.20 µM. Compound **1** showed antifungal activity against *F. fujikuroi* with the inhibition zone radius of 6.5 mm at the concentration of 100 µM (Figure 8). Compounds **7**, **10**, and **11** exhibited antifungal activities against *D. citri*, with the inhibition zone radius of 4.7 mm, 5.3 mm, and 14 mm, respectively, at the concentration of 100 µM (Figure 8). Prochloraz was used as the positive control and showed an inhibition zone radius of 17 mm and 15 mm against *F. fujikuroi* and *D. citri* at the concentration of 100 µM, respectively (Figure 8).

## 3. Materials and Methods

### 3.1. General Experimental Procedures

The UHPLC-MS/MS spectrum was obtained on a high-resolution Q-TOF mass spectrometry Bruker impactHD (Bruker, Switzerland, Germany), combined with Ultimate3000 UHPLC (Thermo Fisher Scientific, Waltham, MA, USA). A Thermo Scientific LTQ Orbitrap XL spectrometer (Thermo Fisher Scientific, Bremen, Germany) was used to measure HR-ESI-MS. Implen Gmbh NanoPhotometer N50 Touch (Implen, Munich, Germany) was used to record the UV spectrum. Nicoiet 380 (Thermo Fisher, Waltham, MA, USA) was used to measure the IR spectrum. Optical rotations were measured on a JASCO P-1020 digital polarimeter (JASCO, Tokyo, Japan). NMR spectra were measured on JEOL JNM-ECZ400S (JEOL, Tokyo, Japan). The Waters 1525 system was used for HPLC purification. Silica gel (200–300 mesh) was employed for chromatographic separation. Thin-layer chromatography was recorded on precoated silica gel GF254 plates.

### 3.2. Actinomycic Materials

The actinomycetes *Nocardiopsis* sp. LX-1 was isolated from the Antarctic krill *Euphausia superba* provided by Qingdao Dongfeng Ocean Fishing Co. LTD in 2019. The strain was deposited in the State Key Laboratory of Microbial Technology, Institute of Microbial Technology, Shandong University, Qingdao, China.

The identification of the actinomycetes LX-1 was determined by the analysis of the 16S rDNA gene sequence in NCBI (Nucleotide BLAST: Search nucleotide databases using a nucleotide query (nih.gov)). The 16S rDNA gene sequence of LX-1 was obtained through the polymerase chain reaction (PCR) method. The fresh actinomycetes LX-1 (about 1.00 mg) was dispersed in a 50-μL lysis buffer (Takara, Cat# 9164) and then was saved in a metal bath (Yooning, Hangzhou, China) at 100 °C for 30 min to extract its genomic DNA as the template DNA in PCR. The PCR was conducted in a final volume of 50 μL, which was composed of the template DNA (3 μL), primers 27F (1 μL) and 1492R (1 μL), PrimeSTAR^®^ Max DNA Polymerase (25 μL, Takara, Cat# R045A), and ultrapure water (20 μL), under the following procedures: (1) initial denaturation at 98 °C for 5 min, (2) denaturation at 98 °C for 30 s, (3) annealing at 55 °C for 30 s, (4) extension at 72 °C for 1.5 min, and (5) final extension at 72 °C for 10 min. Steps (2)–(4) were repeated 35 times. The PCR product was submitted to BGI Genomics for sequencing (BGI, Qingdao, China). The sequence of LX-1 was searched in the NCBI nucleotide collection database through the BLAST program (Nucleotide BLAST: Search nucleotide databases using a nucleotide query (nih.gov)). The actinomycetes LX-1 was identified as *Nocardiopsis* sp. whose 969 base pair 16S sequence had 99.9% sequence identity to that of *Nocardiopsis* sp. E251 (MT533941). The sequence data have been submitted to GenBank with accession number OL687477.

### 3.3. Molecular Networking

#### 3.3.1. UHPLC Parameters

The HPLC C_18_ column (Hitachi, Tokyo, Japan, 250 mm × 4.6 mm, 5 µm) was used for liquid chromatography with the running temperature of 30 °C. Compounds were searched by UV-detector PDA with a wavelength from 190 to 400 nm, and the detection wavelengths of 210 and 254 nm were recorded to characterize the peaks. The mobile phases were MeOH (A)/H_2_O (B). The elution gradient program (time (min), %A) was (0.00, 5), (5.00, 5), (60.00, 100), (75.00, 100), (80.00, 5), and (90.00, 5). The volume of the sample was 20 µL in each injection with 1.00 mL/min flow velocity.

#### 3.3.2. MS/MS Parameters

MS/MS analyses were performed by high-resolution Q-TOF mass spectrometry by using a Bruker impactHD. The ESI source parameters were set as follows: capillary source voltage at 3500 V, positive-ion mode, drying-gas temperature at 200 °C, drying-gas flow rate at 4 L/min, and end plate offset voltage at 500 V. MS scans were recorded in full scan mode with a range of *m/z* 50−1500 (100 ms scan time), and the mass resolution was 40,000 at *m/z* 1222.

#### 3.3.3. Molecular Network Analysis

The molecular network was formed by GNPS workflow (http://gnps.ucsd.edu, accessed on 29 November 2021) [18]. Bruker Daltonics was used to convert the UHPLC-MS/MS raw data file into .mzXML. The parameter settings of the molecular network were detailed in our previous research [22,23]. The results were visualized by using the software package Cytoscape 3.8.0 (Download from https://cytoscape.org/).

### 3.4. Extraction and Isolation

The actinomycetes *Nocardiopsis* sp. LX-1 was cultured in a NB liquid medium in 100 Erlenmeyer flasks (200 mL in each 500 mL flask) at 20 °C for 45 days. The mycelia were filtered from the broth by two layers of gauze. Then, the mycelia were first extracted by ethyl acetate (EA) three times (3 × 200 mL) and then with dichloromethane (DCM)/methanol (MA) (*v/v*, 1:1) three times (3 × 200 mL). The c broth was obtained through repeated extraction with EA (3 × 20 L). All of the fungal crude extracts were put together and evaporated to dryness under reduced pressure to provide a residue (2.2 g). The residue was subjected to silica gel column chromatography (CC) eluted with EA–petroleum ether (PE) (0–100%) and MA–EA (0–100%) to obtain four fractions (Fr.1–Fr.4). Fr.2 was the pure compound **12** (70.5 mg). Fr.3 was separated through CC on silica gel eluted with EA–PE (0–50%) to give three fractions (Fr.3.1–Fr.3.3). Fr.3.2 was purified by using semipreparative HPLC on an ODS column (Kromasil C_18_, 250 × 10 mm, 5 µm, 2 mL/min) eluted with 50% MA–H_2_O to give compounds **2** (1.3 mg) and **9** (1.0 mg). Fr.3.3 was separated on HPLC eluted with 45% MA–H_2_O for **11** (1.9 mg). Fr.4 was separated through CC on silica gel eluted with EA–PE (30–90%) to afford three fractions (Fr.4.1–Fr.4.3). Fr.4.2 was subjected on HPLC with 35% MA–H_2_O for **3** (17.4 mg), **4** (17.9 mg), **6** (19.9 mg), and **10** (4.2 mg). Fr.4.3 was separated by HPLC eluted with 30% MA–H_2_O to give four fractions (Fr.4.3.1–Fr.4.3.4). Fr.4.3.4 was the pure compound **7** (44.4 mg). Fr.4.3.2 was further purified through HPLC eluted with 25% MA–H_2_O for **8** (36.8 mg). Fr.4.3.3 was further purified through HPLC eluted with 25% MA–H_2_O for **1** (1.6 mg) and **5** (13.6 mg). The details are as follows.

Nocarpyrroline A (**1**): amorphous white powder; UV (CH_3_OH) λ_max_ (log *ε*): 288 (2.85); [α${]}_{D}^{20}$ −129.13° (c 0.033, MeOH); ^1^H and ^13^C NMR data (see Table 2); HR-ESI-MS *m/z* [M − H]^−^ 257.0939 (calculated for C_14_H_13_O_3_N_2_, 257.0932) and [M + H]^+^ peak at *m/z* 259.1080 (calculated for C_14_H_15_O_3_N_2_, 259.1077).

Daidzene (**2**): amorphous white powder; ^1^H and ^13^C NMR data (see Appendix A); HR-ESI-MS *m/z* [M − H]^−^ 253.0501 (calculated for C_15_H_9_O_4_, 253.0506) and [M + H]^+^ 255.0660 (calculated for C_15_H_11_O_4_, 255.0652).

Cyclo(d-Pro-l-Val) (**3**): amorphous white powder; ^1^H and ^13^C NMR data (see Appendix A); specific OR data (see Appendix A); HR-ESI-MS *m/z* [M − H]^−^ 195.1130 (calculated for C_10_H_15_O_2_N_2_, 195.1139) and [M + H]^+^ 197.1294 (calculated for C_10_H_17_O_2_N_2_, 197.1285).

Cyclo(4-methyl-d-Pro-l-Nva) (**4**): amorphous white powder; ^1^H and ^13^C NMR data (see Appendix A); specific OR data (see Appendix A); HR-ESI-MS *m/z* [M − H]^−^ 209.1288 (calculated for C_11_H_17_O_2_N_2_, 209.1296) and [M + H]^+^ 211.1449 (calculated for C_11_H_19_O_2_N_2_, 211.1441).

Cyclo(*trans*-4-OH-l-Pro-l-Phe) (**5**): colorless oil; ^1^H and ^13^C NMR data (see Appendix A); specific OR data (see Appendix A); HR-ESI-MS *m/z* [M − H]^−^ 259.1083 (calculated for C_14_H_15_O_3_N_2_, 259.1088), and [M + H]^+^ 261.1244 (calculated for C_14_H_17_O_3_N_2_, 261.1234) and 283.1063 (calculated for C_14_H_16_O_3_N_2_Na, 283.1053).

Cyclo(l-Pro-l-Phe) (**6**): colorless oil; ^1^H and ^13^C NMR data (see Appendix A); specific OR data (see Appendix A); HR-ESI-MS *m/z* [M − H]^−^ 243.1133 (calculated for C_14_H_15_O_2_N_2_, 243.1139), and [M + H]^+^ 245.1294 (calculated for C_14_H_17_O_2_N_2_, 245.1285).

Cyclo(d-Pro-l-Phe) (**7**): colorless oil; ^1^H and ^13^C NMR data (see Appendix A); specific OR data (see Appendix A); HR-ESI-MS *m/z* [M − H]^−^ 243.1133 (calculated for C_14_H_15_O_2_N_2_, 243.1139), and [M + H]^+^ 245.1293 (calculated for C_14_H_17_O_2_N_2_, 245.1285) and 267.1112 (calculated for C_14_H_16_O_2_N_2_Na, 267.1104).

Cyclo(4-hydroxyl-l-Pro-l-Leu) (**8**): amorphous white powder; ^1^H and ^13^C NMR data (see Appendix A); specific OR data (see Appendix A); HR-ESI-MS *m/z* [M − H]^−^ 225.1238 (calculated for C_11_H_17_O_3_N_2_, 225.1245), and [M + H]^+^ 227.1398 (calculated for C_11_H_19_O_3_N_2_, 227.1390) and 249.1217 (calculated for C_11_H_18_O_3_N_2_Na, 249.1210).

2,4(1*H*, 3*H*)Quinazolinedione (**9**): amorphous white powder; ^1^H and ^13^C NMR data (see Appendix A); HR-ESI-MS m/z [M − H]^−^ 161.0345 (calculated for C_8_H_5_O_2_N_2_, 161.0357).

Uracil (**10**): colorless needles; ^1^H and ^13^C NMR data (see Appendix A); HR-ESI-MS *m/z* [M − H]^−^ 111.0187 (calculated for C_4_H_3_O_2_N_2_, 111.0200), and [M + H]^+^ 113.0351 (calculated for C_4_H_5_O_2_N_2_, 113.0346).

Salvinin A (**11**): pale yellow powder; ^1^H and ^13^C NMR data (see Appendix A); HR-APCI-MS *m/z* [M + H]^+^ 161.0602 (calculated for C_10_H_9_O_2_, 161.0597).

2-Aminobenzoic acid (**12**): pale yellow powder; ^1^H and ^13^C NMR data (see Appendix A); HR-ESI-MS *m/z* [M − H]^−^ 136.0393 (calculated for C_7_H_6_O_2_N, 136.0404), and [M + H]^+^ 138.0554 (calculated for C_7_H_8_O_2_N, 138.0550).

### 3.5. Antibacterial Activity Assay

The antibacterial activities were evaluated by the conventional broth dilution assay [45]. Five phytopathogenic bacteria (*Xanthomonas citri pv. malvacearum*, *X. axonopodis*, *Comamonas terrigena*, *Pseudomonas syringae*, and *Dickeya chrysanthemi*), four animal pathogenic bacteria (*Escherichia coli*, *P. aeruginosa*, *Staphylococcus aureus*, and *Bacillus subtilis*), and eight marine fouling bacteria (*Aeromonas hydrophila*, *A. salmonicida*, *Enterobacter cloacae*, *P. fulva*, *Vibrio anguillarum*, *V. harveyi*, *Photobacterium halotolerans*, and *P. angustum*) were used, and cipofloxacin (CPFX) and DMSO were used as the positive and negative control, respectively. The antibacterial activity assay was carried out by using previously described methods [22,23]. The tested concentrations of isolated compounds and CPFX were 100 µM, 50 µM, 25 µM, 12.5 µM, 6.25 µM, 3.13 µM, 1.56 µM, 0.78 µM, 0.39 µM, and 100 µM, 50 µM, 25 µM, 12.5 µM, 6.25 µM, 3.13 µM, 1.56 µM, 0.78 µM, 0.39 µM, 0.20 µM, 0.10 µM, 0.049 µM, and 0.024 µM, respectively.

### 3.6. Antifungal Activity Assay

The antifungal activity against *Candida albicans* was evaluated through the conventional broth dilution method [45]. CPFX and DMSO were used as the positive and negative control, respectively. The antifungal activities against eight phytopathogenic fungi, *Aspergillus niger*, *Alternaria alternata*, *Diaporthe citri*, *Fusarium fujikuroi*, *F. oxysporum*, *F. graminearum*, *F. proliferatum*, and *Colletotrichum* sp., were assessed through the modified agar diffusion test method [46]. The isolated compounds to be tested were dissolved in acetone at a final concentration of 100 µM. The compound solving solution was transferred to a sterile filter disk (diameter 6 mm, each 20 µL), which was placed on the agar growth medium for the tested fungi. Prochloraz was used as the positive control with the test concentration of 100 µM. Acetone was used as negative control.

## 4. Conclusions

In summary, 16 compounds **a**–**p** were recognized from the metabolites of Antarctic krill (*E. superba*)-derived actinomycetes *Nocardiopsis* sp. LX-1 by the method of molecular networking. One new pyrroline, nocarpyrroline A (**1**), along with 11 known compounds **2**–**12**, were isolated from the actinomycetes *Nocardiopsis* sp. LX-1 according to the molecular networking analysis. Among them, compound **2** was the same as the molecular networking investigated isoflavonoid **e**, and this is the first time isoflavonoid from the genus *Nocardiopsis* has been isolated. New compound **1** showed antibacterial activity against *A. hydrophila*, and antifungal activity against *F. fujikuroi*. Compound **2** exhibited broad-spectrum antibacterial activities against *A. hydrophila*, *D. chrysanthemi*, *C. terrigena*, and *X. citri pv. malvacearum*, and antifungal activity against *C. albicans*. Compounds **7** and **10** displayed antibacterial activities against *A. hydrophila*, and **7**, **10**, and **11** revealed antifungal activities against *D. citri*. None of the annotated compounds **a**–**p** by the method of molecular networking had been isolated from the genus *Nocardiopsis*. Nocarpyrroline A (**1**), features an unprecedented 4,5-dihydro-pyrrole-2-carbonitrile substructure, and it is the first pyrroline discovered from the genus *Nocardiopsis*. This study further demonstrated the potential of Antarctic microbes to produce new bioactive natural products and proved the significance of molecular networking in the research of microbial secondary metabolites.

The extremely cold, arid, and fierce solar radiational environments of Antarctica, have created a unique ecological system containing abundant microbial resources with the ability to produce structurally specific active substances. Antarctic krill, as the foundation of the Antarctic marine ecosystem, contain rich symbiotic or parasitical microorganisms that produce special secondary metabolites. Few research papers have studied Antarctic krill-derived microorganisms and their secondary metabolites until now. More attention should be paid to chemical investigation and bioactive evaluation of the natural products isolated from Antarctic krill-derived microorganisms, which could find new bioactive compounds to provide the structural basis for new drug development.

## Figures and Tables

**Figure 1 marinedrugs-21-00127-f001:**
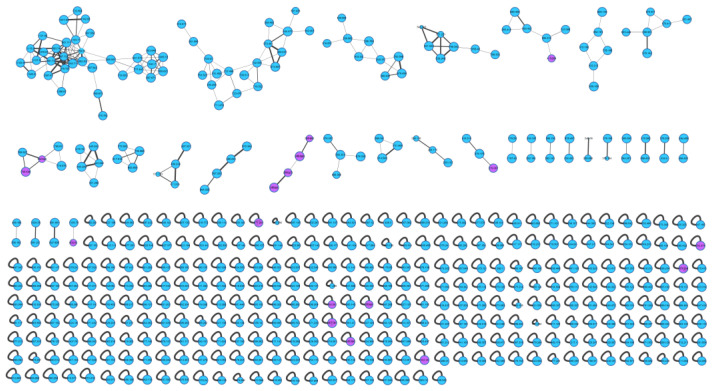
Molecular network of the actinomycetes *Nocardiopsis* sp. LX-1. The purple nodes are compounds annotated by molecular networking and blue nodes are compounds not identified by molecular networking. The numbers in nodes mean precursor mass of compounds. The size of the nodes decided by precursor intensity represents the quantity of the compounds.

**Figure 2 marinedrugs-21-00127-f002:**
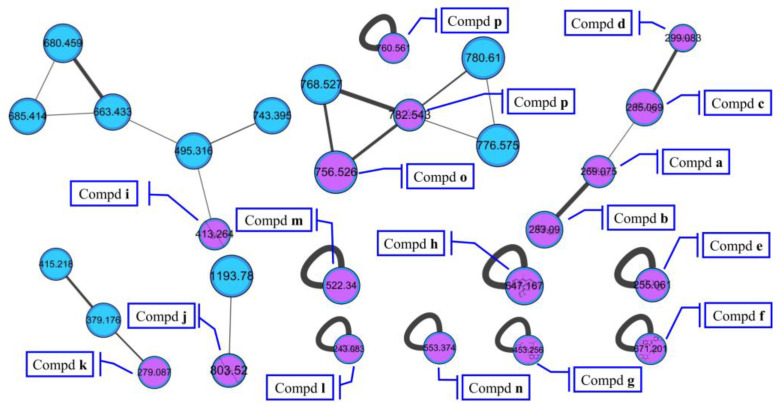
Compounds **a**–**p** identified by molecular networking from *Nocardiopsis* sp. LX-1. The purple nodes are compounds annotated by molecular networking and blue nodes are compounds not identified by molecular networking. The numbers in nodes mean precursor mass of compounds. The letters **a**–**p** in bold format represent different recognized compounds. The size of the nodes decided by precursor intensity represents the quantity of the compounds.

**Figure 3 marinedrugs-21-00127-f003:**
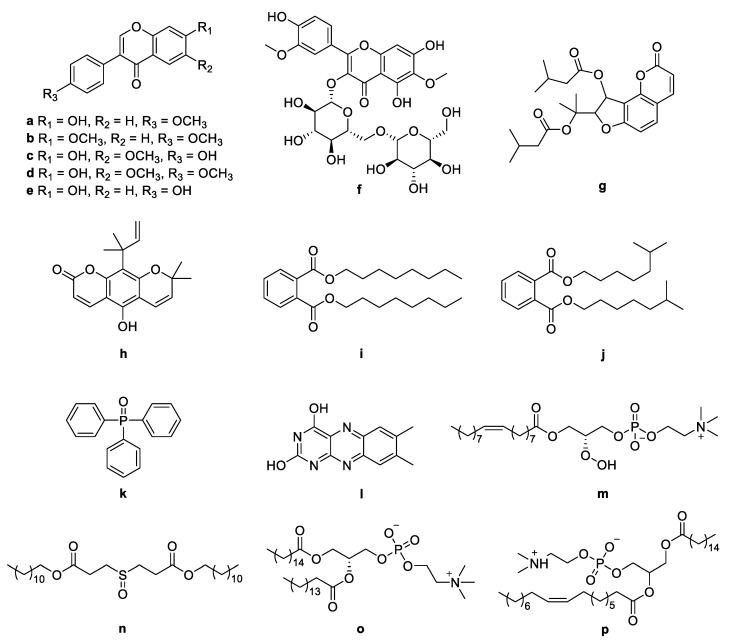
Chemical structures of **a**–**p** identified by molecular networking from *Nocardiopsis* sp. LX-1.

**Figure 4 marinedrugs-21-00127-f004:**
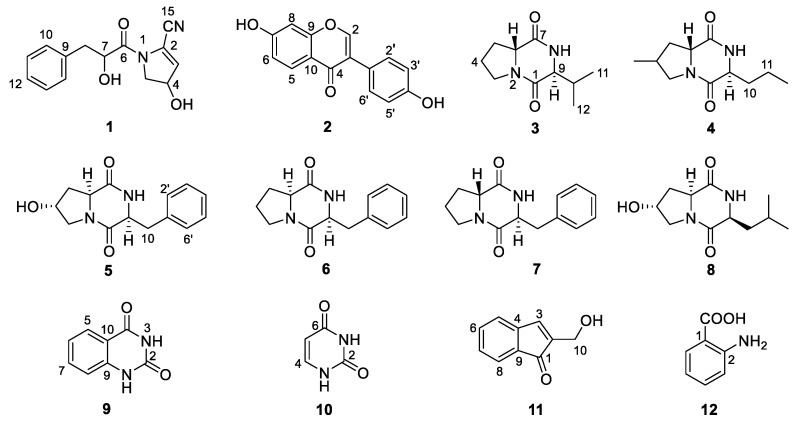
Compounds **1**–**12** isolated from *Nocardiopsis* sp. LX-1.

**Figure 5 marinedrugs-21-00127-f005:**
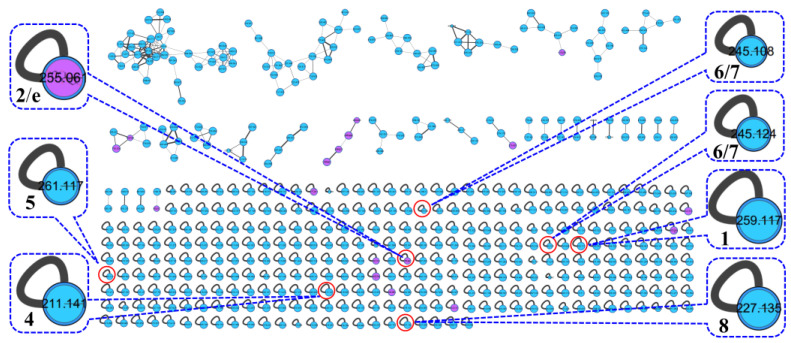
Isolated compounds in the molecular network of *Nocardiopsis* sp. LX-1. The size of the nodes decided by precursor intensity represents the quantity of the compounds.

**Figure 6 marinedrugs-21-00127-f006:**
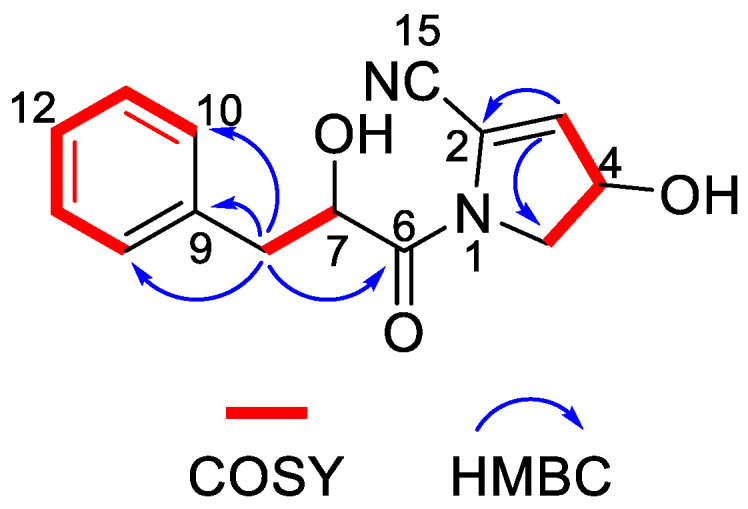
Key COSY and HMBC correlations of nocarpyrroline A (**1**).

**Figure 7 marinedrugs-21-00127-f007:**
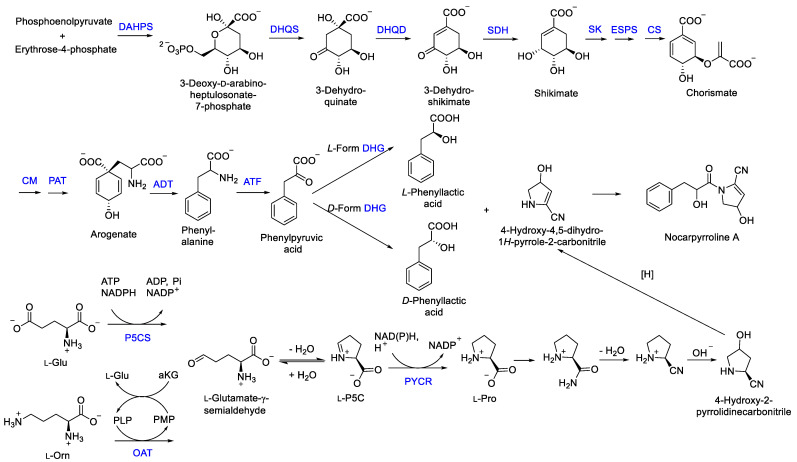
Plausible biogenetic pathway proposed for compound **1** [29,30,31]. DAHPS, 3-deoxy-7-phosphoheptulonate synthase; DHQD, 3-dehydroquinic acid dehydratase; DHQS, 3-dehydroquinic acid synthase; SDH, shikimic acid 5-dehydrogenase; SK, shikimic acid kinase; ESPS, 3-enolpyruvylshikimic acid 5-phosphate synthase; CS, chorismic acid synthase; CM, chorismic acid mutase; PAT, prephenic acid aminotransferase; ADT, arogenic acid dehydratase; ATF, aminotransferase; DHG, dehydrogenase; P5CS, Δ^1^-pyrroline-5-carboxylate synthase; OAT, ornithine-*δ*-aminotransferase; PYCR, Δ^1^-pyrroline-5-carboxylate reductase.

**Figure 8 marinedrugs-21-00127-f008:**
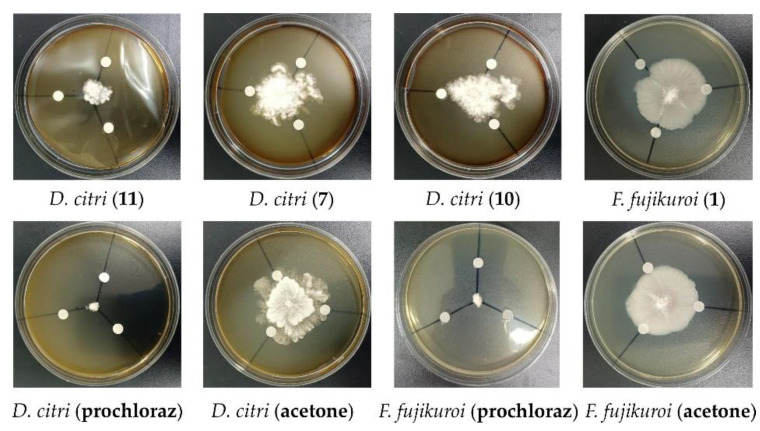
Antifungal activity of compounds **1**, **7**, **10**, and **11**.

**Table 1 marinedrugs-21-00127-t001:** Compounds **a**–**p** identified by molecular networking from *Nocardiopsis* sp. LX-1.

No.	Name	Adduct	Precursor Mass	Exact Mass	CAS Number	RT (min)	Molecular Formula	Class
**a**	Formonentin	[M + H]^+^	269.075	268.074	485723	45.6	C_16_H_12_O_4_	Isoflavonoids
**b**	Dimethoxydaidzein	[M + H]^+^	283.090	282.089	1157397	49.1	C_17_H_14_O_4_	Isoflavonoids
**c**	Glycitein	[M + H]^+^	285.069	284.068	40957833	39.9	C_16_H_12_O_5_	Isoflavonoids
**d**	Afrormosin	[M + H]^+^	299.083	298.29	550798	46.5	C_17_H_14_O_5_	Isoflavonoids
**e**	Daidzein	[M + H]^+^	255.061	254.058	486668	38.9	C_15_H_10_O_4_	Isoflavonoids
**f**	Spinacetin 3-*O*-*β*-gentiobioside	[M + H]^+^	671.201	670.175	101021298	64.0	C_29_H_34_O_18_	Flavonoids
**g**	Athamantin (6CI,7CI)	[M + Na]^+^	453.256	430.49	1892564	39.1	C_24_H_30_O_7_	Coumarin derivatives
**h**	Nordentatin	[2M + Na]^+^	647.167	312.137	1083193150	64.0	C_19_H_20_O_4_	Coumarin derivatives
**i**	Dioctyl phthalate	[M + Na]^+^	413.264	390.277	117840	63.8	C_24_H_38_O_4_	Benzene derivatives
**j**	Diisooctyl phthalate	[2M + Na]^+^	803.520	390.277	131204	63.7	C_24_H_38_O_4_	Benzene derivatives
**k**	Triphenylphosphine oxide	[M + H]^+^	279.087	278.28	791286	45.3	C_15_H_15_OP	Benzene derivatives
**l**	Lumichrome	[M + H]^+^	243.083	242.238	1086802	22.7	C_12_H_10_N_4_O_2_	Pteridine derivatives
**m**	1-(9*Z*-Octadecenoyl)-sn-glycero-3-phosphocholine	[M + H ^−^ O]^+^	522.340	537.667	19420565	65.5	C_26_H_52_NO_8_P	Lipids
**n**	Didodecyl 3,3′-sulfinyldipropionate	[M + Na]^+^	553.374	530.844	17243140	67.2	C_30_H_58_O_5_S	Lipids
**o**	1,2-Dipalmitoyl-l-lecithin	[M + Na]^+^	756.526	734.039	63898	75.9	C_40_H_80_NO_8_P	Lipids
**p**	1-Palmitoyl-2-oleoyl-l-*α*-lecithin	[M + H]^+^	760.561	759.578	26853316	63.8	C_42_H_82_NO_8_P	Lipids
		[M + Na]^+^	782.543					

**Table 2 marinedrugs-21-00127-t002:** NMR spectroscopic data (400/100 MHz) of nocarpyrroline A (**1**) in methanol-*d*_4_.

Position	*δ* _C_	*δ*_H_ (*J* in Hz)
2	135.2, C	
3	119.6, CH	5.73, d (2.7)
4	70.2, CH	4.71, ddd (8.2, 3.5, 2.7)
5	54.5, CH_2_	3.72, dd (13.6, 8.2); 3.62, dd (13.6, 3.5)
6	164.2, C	
7	59.1, CH	4.48, t (4.0)
8	41.1, CH_2_	3.24, d (3.8); 2.96, dd (13.6, 4.5)
9	135.6, C	
10	131.0, CH	7.09–7.04, m
11	129.3, CH	7.24–7.20, m
12	128.2, CH	7.20–7.17, m
13	129.3, CH	7.24–7.20, m
14	131.0, CH	7.09–7.04, m
15	119.7, C	

## Data Availability

The datasets presented in this study can be found in online repositories. The names of the repository/repositories and accession number can be found below: https://www.ncbi.nlm.nih.gov/nuccore/OL687477.1/, accessed on 7 December 2021. The molecular network of the secondary metabolic profile of the actinomycetes *Nocardiopsis* sp. LX-1 can be found in https://gnps.ucsd.edu/ProteoSAFe/status.jsp?task=37dd96194c924f6d9daeef62672ba930, assessed on 29 November 2021.

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
