# Peer review of "New Pyrroline Isolated from Antarctic Krill-Derived Actinomycetes Nocardiopsis sp. LX-1 Combining with Molecular Networking"

_marinedrugs, 2023, doi:10.3390/md21020127_

Round 1

Reviewer 1 Report

The following MS can not be published in its current state. The following issues should be addressed.

-English editing is required, there are many grammatical and typing mistakes.

-The title should be modified to ``New Pyrroline from the Antarctic Krill Derived Actinomycetes Nocardiopsis sp. LX-1 Combined with Molecular Networking``

-Avoid the use of novel, so change it to new.

-Lines from 18 to 21, need rephrasing.

-The type of assay and the results of positive control should be added in the abstract.

-The family of the microbial genus and kirll should be added in the abstract.

_nocarpyrroline A, should be added to the keywords.

- the significance of discovering new antimicrobial agents should be highlighted in the introduction.

- English should be checked and revised throughout the whole MS. The proper tenses for verbs should be used. Canidia albicans, this is incorrect.

- Under figures 1 and 2, an explanation of the blue and purple color should be added in the legends. Also, letters from be clarified.

In figure 4. The structure of compound 1 should be redrawn, the atoms are sticking also the numbers.

-Evidence that supports the existence of the cyano group should be included. IR data should be added and discussed also, how the authors located the position of cyano group.

-Authors should explain, why they selected to test the antimicrobial activity of the isolated compounds.

- Identification of the bacteria should be added.

-The tested concentration of the compounds and control should be added.

-Suggestions and future perspectives should be added to the conclusion.

-The suggested biosynthetic pathway for compound 1, should be included and discussed.

Author Response

Thanks very much for your effort for our manuscript. We have revised our manuscript carefully according to your comments. The changes in our new version of manuscript were highlighted in red.

Question 1:

-English editing is required, there are many grammatical and typing mistakes.

Answer 1:

Thanks very much for your kind comment. The whole English of our manuscript has been edited by MDPI editing service.

Question 2:

-The title should be modified to "New Pyrroline from the Antarctic Krill Derived Actinomycetes Nocardiopsis sp. LX-1 Combined with Molecular Networking".

Answer 2:

Thanks very much for your kind comment. The title has been changed to "New Pyrroline isolated from Antarctic Krill-Derived Actinomycetes Nocardiopsis sp. LX-1 Combining with Molecular Networking".

Question 3:

-Avoid the use of novel, so change it to new.

Answer 3:

Thanks very much for your kind comment. The novel has been changed to new.

Question 4:

-Lines from 18 to 21, need rephrasing.

Answer 4:

Thanks very much for your kind comment. The sentences have been rephrased in lines 19–23.

Question 5:

-The type of assay and the results of positive control should be added in the abstract.

Answer 5:

Thanks very much for your kind comment. The type of assay and the results of positive control have been added in the abstract.

Question 6:

-The family of the microbial genus and krill should be added in the abstract.

Answer 6:

Thanks very much for your kind comment. The family of the microbial genus and krill have been added in the abstract.

Question 7:

_nocarpyrroline A, should be added to the keywords.

Answer 7:

Thanks very much for your kind suggestion. Nocarpyrroline A has been added to the keywords.

Question 8:

- the significance of discovering new antimicrobial agents should be highlighted in the introduction.

Answer 8:

Thanks very much for your kind suggestion. The significance of discovering new antimicrobial agents has been highlighted in the introduction.

Question 9:

- English should be checked and revised throughout the whole MS. The proper tenses for verbs should be used. Canidia albicans, this is incorrect.

Answer 9:

Thanks very much for your kind remind. English has been checked and revised throughout the whole MS using the MDPI English editing service. The proper tenses for verbs have been used. The wrong words “Canidia albicans” have been corrected into “Candida albicans”.

Question 10:

- Under figures 1 and 2, an explanation of the blue and purple color should be added in the legends. Also, letters from be clarified.

Answer 10:

Thanks very much for your kind suggestion. The blue and purple color, and letters in figures 1 and 2 have been clarified in their legends.

Question 11:

In figure 4. The structure of compound 1 should be redrawn, the atoms are sticking also the numbers.

Answer 11:

Thanks very much for your kind comment. The structure of compound 1 has been redrawn.

Question 12:

-Evidence that supports the existence of the cyano group should be included. IR data should be added and discussed also, how the authors located the position of cyano group.

Answer 12:

Thanks very much for your kind comment. The IR absorption band at 2253 cm1 (Figure S9), combined with the molecular formula of C14H14O3N2 and the nine degrees of unsaturation, suggested that there was one cyano group in 1. The high field shift of C-2 and low field shift of C-3 indicated that the cyano group was linked at C-2.

Question 13:

-Authors should explain, why they selected to test the antimicrobial activity of the isolated compounds.

Answer 13:

Thanks very much for your kind comment. All the actinomycetes Nocardiopsis sp. LX-1 isolated compounds (112) were evaluated for their antimicrobial activities for the fermentation broth crude extracts of the actinomycetes LX-1 displayed antimicrobial activity against C. albicans with the inhibition of 70±0.46% at the concentration of 100 µg/mL.

Question 14:

- Identification of the bacteria should be added.

Answer 14:

Thanks very much for your kind comment. The identification of the bacteria has been supplied in section 3.2. Actinomycic Materials.

Question 15:

-The tested concentration of the compounds and control should be added.

Answer 15:

Thanks very much for your kind suggestion. The tested concentration of the compounds and control has been added in sections 3.5. Antibacterial Activity Assay and 3.6. Antifungal Activity Assay.

Question 16:

-Suggestions and future perspectives should be added to the conclusion.

Answer 16:

Thanks very much for your kind suggestion. Suggestions and future perspectives have been added to the conclusion.

Question 17:

-The suggested biosynthetic pathway for compound 1, should be included and discussed.

Answer 17:

Thanks very much for your kind suggestion. The suggested biosynthetic pathway for compound 1 has been supplied in section 2.3. Plausible biogenetic pathway proposed for compound 1.

Reviewer 2 Report

The following comments should be expalined 

I think that the authors should clear report that all the investigation compounds by Molecular Network Analysis are preliminary investigated specially that only one compounds of them has been isolated.

I think that the NOESY spectrum not clear enough to identify the correlation between H-4 and H-7.

Author Response

Thanks very much for your effort for our manuscript. We have revised our manuscript carefully according to your comments. The changes in our new version of manuscript were highlighted in red.

Question 1:

I think that the authors should clear report that all the investigation compounds by Molecular Network Analysis are preliminary investigated specially that only one compounds of them has been isolated.

Answer 1:

Thanks very much for your kind suggestion. “All the investigation compounds by molecular network analysis are preliminary investigated specially that only one compounds of them has been isolated” has been clear reported in lines 103, 104 and 120–122.

Question 2:

I think that the NOESY spectrum not clear enough to identify the correlation between H-4 and H-7.

Answer 2:

Thanks very much for your kind remind. The NOESY cross peaks were really not clear enough to identify the relative configuration of 1. We attempted to determine the configuration of 1 by the modified Mosher’s method. Unfortunately, it was failed due to the limited quantity of 1.

Reviewer 3 Report

The manuscript entitled “New Pyrroline from the Antarctic Krill Derived Actinomycetes Nocardiopsis sp. LX-1 Combined with Molecular Networking” described the characterization of one new and eleven known compounds from antarctic krill derived actinomycetes Nocardiopsis sp. In bioassay, several compounds showed antibacterial activity. Unfortunately, this manuscript is below the threshold of publications in Marine Drugs. I suggest authors to submit this manuscript elsewhere.

Molecular networking is powerful in two ways: annotating known compounds in a sample; cluster the related compounds together. I agree with the concept that molecular networking can be used to analyze metabolites and guide the separation. However, authors failed to do so. The analysis of molecular networking is not informative at all. Authors discovered the new compound 1 by detailed chemical investigation not guided isolation. Molecular networking is actually not helping.

Apparently, several compounds identified by molecular networking are plastic compounds (i and j). Please pay attention to the contamination during chemical workups.

Authors should try to determine the absolute configuration of the new compound.

Author Response

Thanks very much for your effort for our manuscript. We have revised our manuscript carefully according to your comments. The changes in our new version of manuscript were highlighted in red.

Question 1:

The manuscript entitled “New Pyrroline from the Antarctic Krill Derived Actinomycetes Nocardiopsis sp. LX-1 Combined with Molecular Networking” described the characterization of one new and eleven known compounds from antarctic krill derived actinomycetes Nocardiopsis sp. In bioassay, several compounds showed antibacterial activity. Unfortunately, this manuscript is below the threshold of publications in Marine Drugs. I suggest authors to submit this manuscript elsewhere.

Answer 1:

Thanks very much for your kind suggestion. After analysis by molecular networking, 16 compounds ap were preliminary investigated from the crude extracts of Nocardiopsis sp. LX-1 and none of the recognized compounds ap had been isolated from the genus Nocardiopsis until now. Chemical investigation of the broth fermentative crude extracts of Nocardiopsis sp. LX-1 was carried out according the molecular networking analysis, and led to the isolation of one new compound, nocarpyrroline A (1), along with 11 known compounds 212. Among them, compound 2 was same as the molecular networking investigated isoflavone e, and this is the first time to isolate isoflavone from the genus Nocardiopsis. Compounds 1, 2, 7, 10, and 11 showed antimicrobial activities. The plausible biogenetic pathway of new compound 1 was proposed in section 2.3. I think our manuscript is suitable for Marine Drugs in special issue “Discovering Marine Bioactive Compounds by Molecular Networking”.

Question 2:

Molecular networking is powerful in two ways: annotating known compounds in a sample; cluster the related compounds together. I agree with the concept that molecular networking can be used to analyze metabolites and guide the separation. However, authors failed to do so. The analysis of molecular networking is not informative at all. Authors discovered the new compound 1 by detailed chemical investigation not guided isolation. Molecular networking is actually not helping.

Answer 2:

Thanks very much for your kind suggestion. The molecular networking in our research was used to preliminary investigate the number and the structural types of the secondary metabolites of the actinomycetes Nocardiopsis sp. LX-1 and predicted the possibility to find new compounds to determine the research value of the actinomycetes.

Chemical investigation was carried out to find the compounds annotated by the molecular networking, however, only one annotated compound isoflavone e had been isolated from Nocardiopsis sp. LX-1 and this is the first time to isolate isoflavone from the genus Nocardiopsis.

The new isolated compound 1 was searched in the molecular network, and found it was appeared alone in the molecular network, indicating there is no similar compound with 1 in the metabolic profile of the Nocardiopsis sp. LX-1.

Question 3:

Apparently, several compounds identified by molecular networking are plastic compounds (i and j). Please pay attention to the contamination during chemical workups.

Answer 3:

Thanks very much for your kind remind. The plastic products were avoided during the chemical workups. The plastic compounds might be contaminated by the organic extracted solvent, which should be double distillated to purify.

Question 4:

Authors should try to determine the absolute configuration of the new compound.

Answer 4:

Thanks very much for your kind comment. The absolute configuration of 1 was attempted to determine by the modified Mosher’s method. Unfortunately, it was failed due to the limited quantity of 1.

Round 2

Reviewer 1 Report

No comment

Author Response

Thanks very much for your agreement to publish our manuscript.

Reviewer 3 Report

“…..possibility to find new compounds to determine the research value of the actinomycetes.”

-----Apparently, authors only found several known compound classes. I do not think this is a good sample for determining the research value of the actinomycetes. Also, authors claims they have discovered a very well-known compound (even a class of compounds) from a certain genus (Nocardiopsis sp.) for the first time. For me, this is below the Marine Drugs threshold.

The new isolated compound 1 was searched in the molecular network, and found it was appeared alone in the molecular network, indicating there is no similar compound with 1 in the metabolic profile of the Nocardiopsis sp. LX-1.

------The so-called appeared alone in the molecular network is tricky. GNPS works best when one considers discovering new analogs of a known compound or a certain compound class. This will end up with a very nice cluster. Those unclustered nodes are not necessary chemically potent. All you can tell is there is no compound showing enough MSMS similarity with the single node in this certain sample.

“The plastic compounds might be contaminated by the organic extracted solvent, which should be double distillated to purify.”

------For metabolomics analysis, authors should apply LC-MS grade/HPLC grade solvents to process the samples. The contamination of plastic compounds are not acceptable for metabolomics analysis.

 “Thanks very much for your kind comment. The absolute configuration of 1 was attempted to determine by the modified Mosher’s method. Unfortunately, it was failed due to the limited quantity of 1.”

------For a paper in Marine Drugs, I am expecting a much more solid study in terms of stereochemistry determination.

Author Response

Dear reviewer:

Thanks very much for your effort for our paper entitled "New Pyrroline Isolated from Antarctic Krill-Derived Actinomycetes Nocardiopsis sp. LX-1 Combining with Molecular Networking". We have revised our manuscript carefully according to your comments. The changes in our new version of manuscript were highlighted in red.

Question 1

“…..possibility to find new compounds to determine the research value of the actinomycetes.”

-----Apparently, authors only found several known compound classes. I do not think this is a good sample for determining the research value of the actinomycetes. Also, authors claims they have discovered a very well-known compound (even a class of compounds) from a certain genus (Nocardiopsis sp.) for the first time. For me, this is below the Marine Drugs threshold.

Answer 1

Thanks very much for your kind comment. Besides of the analysis of the secondary metabolic profile of Nocardiopsis sp. LX-1 by molecular networking and the isolation of the actinomycetes crude extract, the plausible biogenetic pathway of new compound 1 was proposed in section 2.3, the isolated compounds have been found in the molecular network of Nocardiopsis sp. LX-1, the NMR data of the isolated compounds have been supplied, the HRMS spectra of the isolated compounds have been measured, and the MS/MS spectra of compound ap compared with GNPS library spectra have been provided in supporting information. I think our manuscript is suitable for Marine Drugs in special issue “Discovering Marine Bioactive Compounds by Molecular Networking”.

Question 2

The new isolated compound 1 was searched in the molecular network, and found it was appeared alone in the molecular network, indicating there is no similar compound with 1 in the metabolic profile of the Nocardiopsis sp. LX-1.

------The so-called appeared alone in the molecular network is tricky. GNPS works best when one considers discovering new analogs of a known compound or a certain compound class. This will end up with a very nice cluster. Those unclustered nodes are not necessary chemically potent. All you can tell is there is no compound showing enough MS/MS similarity with the single node in this certain sample.

Answer 2

Thanks very much for your kind remind. The sentence has been changed into “there is no compound showing enough MS/MS similarity with the single node in this certain sample”.

Question 3

“The plastic compounds might be contaminated by the organic extracted solvent, which should be double distillated to purify.”

------For metabolomics analysis, authors should apply LC-MS grade/HPLC grade solvents to process the samples. The contamination of plastic compounds are not acceptable for metabolomics analysis.

Answer 3

Thanks very much for your kind comment. The sentence has been changed into “The plastic products were avoided during the extraction and purification, and the plasticizers i and j might be contaminated by the organic extracted solvent, which should be used by LC-MS grade/HPLC grade solvents to process the samples”.

Question 4

 “Thanks very much for your kind comment. The absolute configuration of 1 was attempted to determine by the modified Mosher’s method. Unfortunately, it was failed due to the limited quantity of 1.”

------For a paper in Marine Drugs, I am expecting a much more solid study in terms of stereochemistry determination.

Answer 4

Thanks very much for your kind comment. We really expect to determine the configuration of the compound 1, while it was used up through the modified Mosher’s reaction.